# Posterior Reversible Encephalopathy Syndrome after Pazopanib Therapy

**DOI:** 10.3390/diseases11020076

**Published:** 2023-05-23

**Authors:** Madhavkumar Savaliya, Drishty Surati, Ramesh Surati, Shailesh Padmani, Stergios Boussios

**Affiliations:** 1Department of Medical Oncology, Medway NHS Foundation Trust, Windmill Road, Gillingham ME7 5NY, UK; drishty.surati@nhs.net (D.S.); stergiosboussios@gmail.com (S.B.); 2Metas Adventist Hospital, Surat 395001, India; rameshrsurati@yahoo.com; 3Sahyog Imaging Center, Surat 395009, India; padmanishailesh@yahoo.co.in; 4School of Cancer & Pharmaceutical Sciences, Faculty of Life Sciences & Medicine, King’s College London, London SE1 9RT, UK; 5Kent Medway Medical School, University of Kent, Canterbury CT2 7LX, UK; 6AELIA Organization, 9th Km Thessaloniki–Thermi, 57001 Thessaloniki, Greece

**Keywords:** renal cell cancer, tyrosine kinase inhibitors, pazopanib, posterior reversible encephalopathy syndrome

## Abstract

The term posterior reversible encephalopathy syndrome (PRES) refers to an acute syndrome characterised by a range of neurological symptoms and posterior transient changes on neuroimaging. Common clinical presentation includes headache, confusion, visual disturbances, seizures, and focal neurological deficit. With the advancement and increasing availability of neuroimaging, this syndrome is increasingly recognised. There are several underlying causes for PRES, including certain medications. Tyrosine kinase inhibitors (TKIs) such as pazopanib can increase the risk of developing PRES by markedly elevating the blood pressure due to its effect of inhibition of vascular endothelial growth factor receptors (VEGFRs). We are reporting a case of a 55-year-old male patient with the clear cell type of renal cell carcinoma (RCC) who developed PRES within a short period after starting pazopanib therapy. With the effective control of his blood pressure and discontinuation of pazopanib, his typical magnetic resonance imaging (MRI) lesion of PRES resolved in the follow-up scan after four weeks.

## 1. Introduction

Renal cell carcinoma (RCC) is one of the most commonly occurring cancers globally, ranking among the top 10. The number of new cases detected each year surpasses 70,000 in the United States and 350,000 worldwide [1]. The clear cell subtype of RCC comprises more than 70% of all RCC cases and exhibits tumour cells characterised by clear-cytoplasm-containing lipids. Patients who are diagnosed with localised disease have a favorable 5-year survival rate of 73%, whereas those who suffer from metastatic RCC have a less promising prognosis with a median progression-free survival (PFS) of 15.1 months [2].

The pathogenesis of RCC, which is rich in blood vessels, involves the deactivation of the *Von Hippel-Lindau* (*VHL*) gene. This deactivation leads to an increased expression of proteins that promote neovascularization, including the crucial vascular endothelial growth factor (VEGF), the platelet-derived growth factor (PDGF), the transforming growth factor (TGF), and others. A better understanding of the biological and molecular pathogenesis of RCC has led to significant progress in targeted therapy for advanced RCC. This progress has been achieved through the introduction of tyrosine kinase inhibitors (TKIs) that are specifically designed to target the proteins involved in RCC pathogenesis along with the mammalian target of rapamycin (mTOR) inhibitors and immunotherapy administered either as single agents or combined with TKIs [3]. Pazopanib is an oral TKI that targets the VEGF receptor, the PDGF receptor, and receptor encoding by the proto-oncogene c-Kit [4]. It has the ability to halt the movement and growth of both vascular endothelial cells and tumour cells involved in angiogenesis by inhibiting the activity of VEGF, PDGF, and other molecular targets. This makes pazopanib an effective therapeutic tool in fighting solid metastatic lesions as well as pulmonary and soft tissue metastases.

Posterior reversible encephalopathy syndrome (PRES), which was previously referred to as posterior reversible leukoencephalopathy or reversible posterior cerebral oedema syndrome, is a clinical and radiological condition. It is characterised by symptoms such as headaches, confusion, visual disturbances, seizures, and transient changes that appear posteriorly on neuroimaging [5]. From a diagnostic perspective, the main features related to the syndrome—namely the cerebral edema and elevated blood pressure—may be evaluated by using the potential magnetic resonance imaging (MRI) findings on T2-weighted and fluid-attenuated inversion recovery (FLAIR) images and the blood pressure readings, respectively [6]. Some of these features can be treatment-mediated and should be carefully considered in patients who are treated with immunosuppressants and antiangiogenics such as the TKI pazopanib [7]. The prevalence of PRES has been reported in 0.03% of 21 million people in nationwide inpatient sample data from 2016 to 2018, indicating its rarity [8]. It is a potentially reversible adverse event.

We performed a literature search using the keywords “Posterior Reversible Encephalopathy Syndrome”, “Pazopanib” and “Posterior Reversible Leukoencephalopathy Syndrome”, and we found only 12 cases associated with this coexistence. Here, we present a case of a 55-year-old male patient with PRES who was treated with pazopanib for the clear cell type of RCC, bringing the total number of reported cases up to 13 in the frame of the English literature. Our case report adds onto the small body of literature and highlights that a high index of suspicion and early discontinuation of pazopanib may result in the improvement of patients’ symptoms and in the complete resolution of PRES lesions on MRI.

## 2. Case Description

We report a 55-year-old male with a past medical history of type 2 diabetes mellitus without other comorbidities such as systemic hypertension or renal insufficiency. He had a history of tobacco chewing for over a decade that he quit 5 years before his presentation. He did not smoke or consume alcohol and did not have any significant family history of any major diseases. The patient was suspected to have right RCC in his right kidney based on radiological evidence from a performed computerised tomography (CT) scan of the abdomen and pelvis. From a therapeutic perspective, he underwent a right radical nephrectomy. The immunohistochemical findings of the histology specimen were compatible with a clear cell RCC, whereas the tumour extended into the perinephric fat and adrenal gland. One month following the nephrectomy, the patient commenced biological treatment with the TKI pazopanib at a reduced dosage of 400 mg twice daily. Apart from pazopanib, the patient did not receive any other anticancer treatment such as immunotherapy or cytotoxic agents. Seven days after the initiation of the treatment, he began complaining of a headache in the evening and reported three episodes of vomiting, whilst the blood pressure measured at that time was within the range of 150–170 mm mercury (mmHg) systolic and 80–100 mmHg diastolic.

At some point his symptoms persisted overnight at home, and on the following morning he had an episode of generalised tonic clonic seizure followed by altered sensorium. He was immediately brought to the emergency department, where his blood pressure was 200/100 mmHg. His sensorium was altered with a Glasgow Coma Scale of 9/15 points. He did not report fever, visual disturbances or any sensory symptoms. The routine blood reports, including complete blood count and renal and liver function tests, along with routine urine tests did not reveal abnormalities except hyponatraemia (serum Na 127 mEq/L). An HIV test was negative during his pre-operative evaluation before nephrectomy. An urgent brain MRI (Figure 1a,b) with contrast detected diffuse ill-defined asymmetrical cortical and subcortical hyper intensities involving bilateral parieto-occipital lobes. These findings were suggestive of PRES.

The seizure was treated with an anticonvulsant (levetiracetam) and the patient’s blood pressure was acutely controlled with intravenous labetalol, whereas later the patient was switched to maintenance antihypertensive treatment with nifedipine. At the same time, pazopanib was discontinued and his systolic blood pressure was strictly controlled between 100 and 130 mmHg. The patient gradually began to clinically improve, and his symptoms were completely resolved on the third day of the admission. Indeed, he had no residual neurological deficit and did not have further episodes of seizure in the following days. Therefore, he was safely discharged from the hospital with oral levetiracetam and oral nifedipine with further follow up on outpatient setting (Figure 2). Upon follow-up after 4 weeks’ time, his blood pressure was fairly under control, and a repeat MRI brain (Figure 1c,d) showed complete resolution of PRES lesions.

## 3. Discussion

RCC comprises the largest proportion of kidney tumours and is histologically categorised into clear cell (70%), papillary (10–15%), chromophobe (5–10%), and collecting duct carcinoma (<1%). At the time of diagnosis, approximately 30% of RCC patients exhibit metastatic disease, while nearly 50% of those who receive curative surgery encounter relapse accompanied by distant metastases. Over the past few years, the prognosis of RCC patients has witnessed substantial enhancement owing to the advent of molecular targeted therapies. These treatment modalities encompass various therapies including small-molecule TKIs such as sorafenib, sunitinib, pazopanib, and axitinib; mTOR inhibitors such as temsirolimus and everolimus; and an antiangiogenic antibody named bevacizumab that is typically administered alongside interferon alpha. Among them, pazopanib stands out as a multi-TKI that can be taken orally and targets kinase receptors crucial for tumour cell angiogenesis and proliferation (such as VEGF receptors, PDGF receptors, and the c-Kit receptor). Consequently, this leads to the inhibition of angiogenesis, impeding cell growth and survival and ultimately preventing tumour progression and propagation. Pazopanib received approval from the United States Food and Drug Administration (US FDA) in October 2009 for treating advanced or metastatic RCC based on encouraging results from a phase III randomised clinical trial [9]. The study demonstrated a significant improvement in PFS when compared to the placebo across the entire study population (median PFS 9.2 vs. 4.2 months; hazard ratio (HR), 0.46; 95% CI, 0.34 to 0.62; *p* < 0.0001), as well as within both the treatment-naive subgroup (median PFS 11.1 vs. 2.8 months; HR, 0.40; 95% CI, 0.27 to 0.60; *p* < 0.0001) and the subgroup of patients previously treated with cytokines (median PFS 7.4 vs. 4.2 months; HR, 0.54; 95% CI, 0.35 to 0.84; *p* < 0.001). However, it is important to note that the inhibition of multiple targets by pazopanib can lead to various adverse events, among which hematologic and hepatic toxicities such as elevated aspartate and alanine aminotransferase levels (as well as bilirubin elevation) are particularly significant and necessitate careful monitoring [10]. However, there is still no definitive evidence to support a relationship between the severity of adverse events and efficacy. Because pazopanib blocks the VEGF receptor, it is associated with hypertension and other cardiovascular complications [11]. Apart from hypertension and hematologic and hepatic toxicities, pazopanib may induce hair depigmentation, diarrhea, nausea, anorexia, and vomiting.

PRES is a condition that manifests as various neurological symptoms that include seizures and hypertensive emergencies. However, diagnosing PRES based on clinical presentation alone can be challenging because these findings lack specificity. On the other hand, MRI findings often provide a characteristic pattern and are a critical component in diagnosing PRES. Lesions typically appear in the posterior white matter and may involve the overlying cortex. They show up as hyperintense on T2-weighted images and usually hypointense or isointense on diffusion-weighted images with an increase in the apparent diffusion coefficient, indicating vasogenic oedema [5]. The availability and advancement of brain imaging techniques, especially MRI, have contributed to the increasing recognition of PRES as a clinical syndrome.

The pathophysiology of the development of PRES is still poorly understood. Three hypotheses have been proposed to explain the pathogenesis of PRES. The first hypothesis suggests that cerebral vasoconstriction causes subsequent infarcts in the brain. The second hypothesis proposes that there is a failure of cerebral autoregulation, which leads to vasogenic edema. The third hypothesis involves endothelial damage with blood–brain barrier disruption, resulting in fluid and protein transudation in the brain [12]. Endothelial dysfunction, which is characterised by an impaired vasodilation phenotype and a proinflammatory state of the endothelium, is an on-target effect of VEGF receptor inhibitors. The vasoconstrictive response to these drugs is related to a reduction in the levels of the vasodilator nitric oxide and an increase in vasoactive peptides such as endothelin and arginine vasopressin (AVP). Recent studies suggested that AVP overstimulation plays a significant role in the development of PRES and its associated symptoms, particularly due to its pathophysiologic role in brain edema formation and its involvement in most PRES etiologies [13]. The preferential involvement of posterior brain regions is likely due to regional heterogeneity of the arterial sympathetic innervation. This innervation is highest in the anterior circulation and decreases with an anterior-to-posterior gradient. Therefore, the occipital lobes and other posterior brain regions are at a relatively higher risk for hydrostatic edema [14]. From the diagnostic perspective, if a patient presents with neurologic symptoms and signs and the medical history suggests hypertension, immunosuppressive agents, TKI inhibitors, and/or kidney disease, a brain MRI is strongly recommended. Treatment of PRES is mainly symptomatic, including an effective management of blood pressure, control of seizures, and management of underlying disease or elimination of causative factors [15]. In most cases, PRES is completely reversible, but in severe forms, it might cause substantial morbidity and even mortality, often due to acute hemorrhage or extensive edema in the posterior fossa that leads to obstructive hydrocephalus or compression of the brainstem [16].

VEGF inhibition led to structural remodelling of the resistance vasculature as evidenced by microvascular rarefaction and a significant increase in peripheral vascular resistance with a subsequent increase in blood pressure [17]. Given that VEGF receptor inhibition has been theorised to play a prominent role in the development of hypertension, it may subsequently mediate PRES. Indeed, significant elevation in blood pressure above the limit of cerebral autoregulation can cause the leaking of blood products into the white matter parenchyma. To prevent this, the National Cancer Institute recommends lowering blood pressure to below 140/90 mmHg during VEGF receptor inhibitor treatment (or below 130/80 mmHg for patients with chronic kidney disease or diabetes) [18].

Apart from the supportive measures for the treatment of PRES, if a targeted biological agent or immunosuppressant drug is identified as the underlying trigger for development of this syndrome, then consideration should be given to either discontinuation or dose modification of the offending agent early in the course of disease [19]. Blood pressure management includes gradual lowering of the blood pressure up to 20–25% in initial few hours with the aim of normalising the mean arterial pressure [20]. As the name suggests, PRES is a benign and reversible condition, but long-term functional impairment and death can occur if not diagnosed early or treated properly [21]. Specifically, pazopanib-induced PRES can be associated with intracerebral haemorrhage, which may lead to permanent neurologic deficits or even death.

The occurrence of PRES as a complication of pazopanib therapy is an uncommon association. Until now, 12 cases of such association had been reported in English literature. Table 1 summarises the list of reported cases in the English literature prior to our report.

## 4. Conclusions

The treatment of PRES involves addressing the inciting aetiology. Patients who are treated with pazopanib should undergo regular follow-up and monitoring of their blood pressure; if they exhibit any symptoms suggestive of PRES, then there should be a low threshold for an MRI of the brain. Overall, it is vital to keep this potential association in mind while treating the patients with pazopanib because prompt recognition and management of PRES is critical to preventing permanent neurological damage and improving patient outcomes.

## Figures and Tables

**Figure 1 diseases-11-00076-f001:**
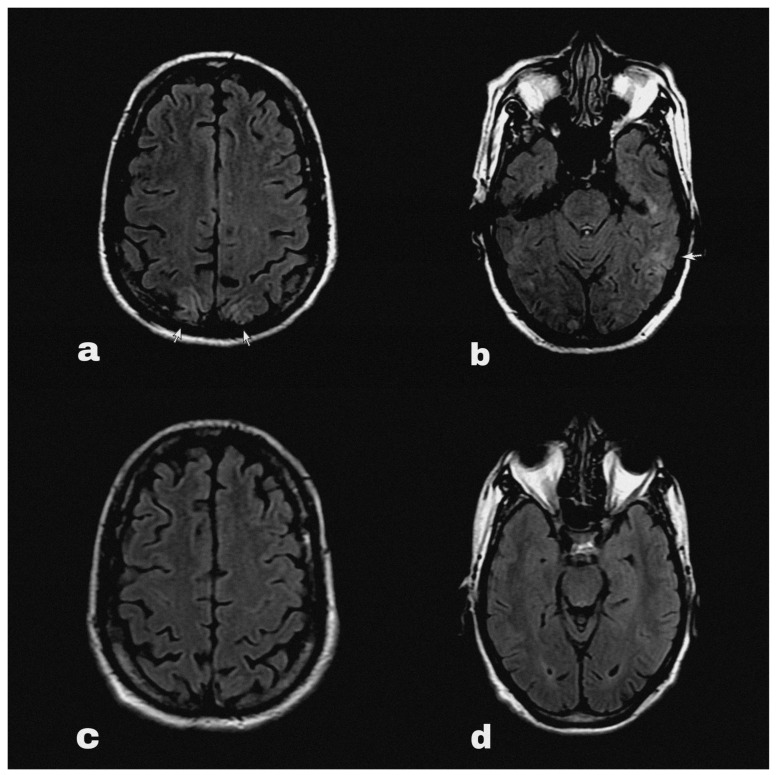
Brain MRI FLAIR sequence showing hyperintensities (marked with arrows) in occipital (**a**) and parietal (**b**) regions at the onset of PRES. The follow-up MRI (**c**,**d**) showed complete resolution of the lesions after 4 weeks.

**Figure 2 diseases-11-00076-f002:**
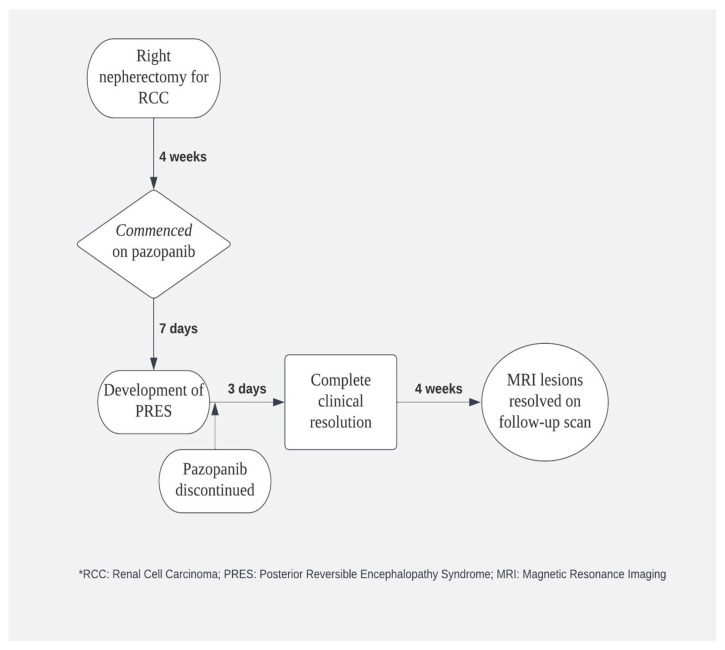
Schematic diagram depicting the timeline of patient’s clinical progression and its association with pazopanib therapy.

**Table 1 diseases-11-00076-t001:** List of reported cases of pazopanib-induced PRES.

Time Interval between Discontinuation of Pazopanib and Resolution of PRES Symptoms	Time Interval between Initiation of Pazopanib and PRES	Dose	Primary Cancer	Age/Gender	Author
1 day	21 days	Not specified	RCC	40/F	Chelis et al. [4]
6 days	8 weeks	800 mg/day	RCC	63/F	Foerster et al. [22]
2 days	4 weeks	Not specified	RCC	76/M	Asaithambi et al. [23]
Not specified	3 weeks	Not specified	RCC	69/F	Miller-Patterson et al. [24]
60 h	16–20 weeks	400 mg/day	Testicular tumour	32/M	Arslan et al. [25]
3 days	9 days	800 mg/day	RCC	56/F	Miaris et al. [26]
5 days	5–6 weeks	800 mg/day	Retroperitoneal soft tissue sarcoma	49/F	Deguchi et al. [27]
5 days	4 days	800 mg/day	HCC	76/F	Wu et al. [28]
5 days	20 months	Not specified	Hurthel cell thyroid carcinoma	56/F	Koleszar et al. [29]
9 days	4 days	Not specified	Uterine sarcoma	64/F	Takahashi et al. [30]
6 days	12 days	600 mg/day	RCC	73/F	Wong So et al. [31]
7 days	3 days	800 mg/day	Uterine sarcoma	64/F	Tatsumichi et al. [32]

Abbreviation: F, Female; M, Male; PRES, posterior reversible encephalopathy syndrome; RCC, renal cell carcinoma; HCC, hepato-cellular carcinoma.

## Data Availability

Not available.

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
