# Peer review of "Posterior Reversible Encephalopathy Syndrome after Pazopanib Therapy"

_diseases, 2023, doi:10.3390/diseases11020076_

Round 1
Reviewer 1 Report
In the manuscript, the authors reported a case of PRES in a patient with RCC following pazopanib therapy. The story looks extremely entertaining, but there are still some doubts.
1. The clinical characteristics of this patient need to be described adequately and not limited to age and sex. At the same time, a patient's family history and lifestyle habits, including smoking history, drinking history, and sleep status, are also of interest.
2. Did the patient use additional chemotherapy drugs after surgery in addition to pazopanib?
3. The description of the onset process of the patient is not sufficiently detailed. When did it start to get worse? Are there any symptoms, such as fever and disturbance of consciousness?
Author Response
Dear Editor and Reviewers,
I am pleased to resubmit for publication the revised version of diseases-2384026 manuscript, entitled “Posterior Reversible Encephalopathy Syndrome after Pazopanib Therapy”.
Thankfully the reviewers provided us with a great deal of guidance, regarding how to better position the article. We are hopeful you agree that this revision will update our invited editorial. All the comments have been addressed, as shown in the revised version of the manuscript, along with this point-by-point response to the reviewers' comments.
All corresponding are blue changes in the manuscript.
Reviewer 1:
General comments:
In the manuscript, the authors reported a case of PRES in a patient with RCC following pazopanib therapy. The story looks extremely entertaining, but there are still some doubts.
Response:
We appreciate you taking the time to offer us your comments and insights related to the paper. Thank you for your positive reinforcement and constructive feedback. We tried to be responsive to your concerns as we approached our revision.
Specific comments:
-
“The clinical characteristics of this patient need to be described adequately and not limited to age and sex. At the same time, a patient's family history and lifestyle habits, including smoking history, drinking history, and sleep status, are also of interest”.
Response: Thank you for your recommendations. We have now amended patients clinical characteristics including age, sex, medical history, habits, family history (Lines 79-82 on revised manuscript).
-
“Did the patient use additional chemotherapy drugs after surgery in addition to pazopanib?”
Response: Thank you for your comment. We have added that patient did not receive any other chemotherapeutic drug after his surgery (Lines 89-90 on revised manuscript).
-
“The description of the onset process of the patient is not sufficiently detailed. When did it start to get worse? Are there any symptoms, such as fever and disturbance of consciousness?”
Response: Thank you for your suggestion. We have now modified patients’ clinical presentation and how it progressed and included the negative history of symptoms. Patient had altered sensorium; we have also mentioned his Glasgow Coma Scale (GCS) score (Lines 95-99 in revised manuscript).
Reviewer 2 Report
The authors have reported a case study of a 55 year male patient undergoing TKI therapy (Pazopanib) for renal metastatic cancer, who underwent a side effect of the drug. The side effect was manifested as posterior reversible encephalopathy syndrome (PRES). The patient started to have headaches, confusion followed by a focal seizure and was brought to hospital. He was also diagnosed with a very high blood pressure. The patient management was done by stopping the Pazopanib, giving anticonvulsants (Levetiracetam) and blood pressure control medicines. The patient reverts to normal as visualized by the MRI after 4 weeks of therapy.
The case study is elaborately reported in the manuscript, with detailed description of the patient management. The discussion is also well explanatory citing the previous literature on PRES after the intake of Pazopanib.
The English language is also very good and describes the findings of the study very clearly.
My suggestions to further improve the manuscript is as follows:
1. Please include a schematic diagram to depict the timeline improvement of the patient.
2. Scheme showing a possible association of Pazopanib with the onset of PRES.
I suggest a minor revision.
Author Response
Dear Editor and Reviewers,
I am pleased to resubmit for publication the revised version of diseases-2384026 manuscript, entitled “Posterior Reversible Encephalopathy Syndrome after Pazopanib Therapy”.
Thankfully the reviewers provided us with a great deal of guidance, regarding how to better position the article. We are hopeful you agree that this revision will update our invited editorial. All the comments have been addressed, as shown in the revised version of the manuscript, along with this point-by-point response to the reviewers' comments.
All corresponding are blue changes in the manuscript.
Reviewer 2:
General comments:
“The authors have reported a case study of a 55 year male patient undergoing TKI therapy (Pazopanib) for renal metastatic cancer, who underwent a side effect of the drug. The side effect was manifested as posterior reversible encephalopathy syndrome (PRES). The patient started to have headaches, confusion followed by a focal seizure and was brought to hospital. He was also diagnosed with a very high blood pressure. The patient management was done by stopping the Pazopanib, giving anticonvulsants (Levetiracetam) and blood pressure control medicines. The patient reverts to normal as visualized by the MRI after 4 weeks of therapy.
The case study is elaborately reported in the manuscript, with detailed description of the patient management. The discussion is also well explanatory citing the previous literature on PRES after the intake of Pazopanib.
The English language is also very good and describes the findings of the study very clearly”.
Response:
Thank you very much for your kind words about our paper. We appreciate the opportunity to revise our work for consideration for publication.
Specific comments:
-
“Please include a schematic diagram to depict the timeline improvement of the patient”.
Response: Thank you for the valuable advice. Schematic diagram showing the timeline of patient’s clinical improvement has been added (Figure 2).
-
“Scheme showing a possible association of Pazopanib with the onset of PRES”.
Response: Thank you for your response. Possible association of pazopanib with Patient’s clinical presentation has been represented in the schematic diagram (Figure 2).
Reviewer 3 Report
Savalyia et al present a case report of a 55 year old male with PRES. This is an interesting and well written case report.
I have only a few minor comments:
-In patients with a disease of the brain, the HIV status is important for initial differential diagnoses. I suggest that this information (is the patient HIV negative or HIV positive?) should be added.
-Has a lumbar puncture (LP) been done? If yes, the results of the LP would be also interesting, i.e to rule out other differential diagnoses
Author Response
Dear Editor and Reviewers,
I am pleased to resubmit for publication the revised version of diseases-2384026 manuscript, entitled “Posterior Reversible Encephalopathy Syndrome after Pazopanib Therapy”.
Thankfully the reviewers provided us with a great deal of guidance, regarding how to better position the article. We are hopeful you agree that this revision will update our invited editorial. All the comments have been addressed, as shown in the revised version of the manuscript, along with this point-by-point response to the reviewers' comments.
All corresponding are blue changes in the manuscript.
Reviewer 3:
General comments:
“Savaliya et al present a case report of a 55 year old male with PRES. This is an interesting and well written case report”.
Response:
We appreciate your valuable feedback and thank you for the positive reinforcement.
Specific comments:
-
“In patients with a disease of the brain, the HIV status is important for initial differential diagnoses. I suggest that this information (is the patient HIV negative or HIV positive?) should be added”.
Response: Thank you for this important suggestion. HIV test was negative during patient’s pre-operative evaluation, which has been added. (Lines 102-103 on revised manuscript)
-
Has a lumbar puncture (LP) been done? If yes, the results of the LP would be also interesting, i.e to rule out other differential diagnoses
Response: Thank you for your comment. Lumbar puncture was not done for this patient.
Round 2
Reviewer 1 Report
My questions have all been resolved.